# How to Create a Geocultural Site's Content–Huta Różaniecka Case Study (SE Poland)

Ewa Skowronek [1,*], Teresa Brzezińska-Wójcik [1] and Waldemar Kociuba [2]

1 Institute of Socio-Economic Geography and Spatial Management, Maria Curie-Skłodowska University, 20-718 Lublin, Poland; teresa.brzezinska-wojcik@mail.umcs.pl
2 Institute of Earth and Environmental Sciences, Maria Curie-Skłodowska University, 20-718 Lublin, Poland; waldemar.kociuba@mail.umcs.pl
* Correspondence: ewa.skowronek2@mail.umcs.pl; Tel.: +48-81-537-68-40

**Abstract:** This study concerns the design of a geocultural site in Huta Różaniecka. It is one of 166 sites prepared for the Kamienny Las na Roztoczu (Roztocze Stone Forest) Geopark project. The site is distinguished, on the one hand, by its interesting geology and geomorphology (exposures of Miocene sea shore with numerous fossils) and, on the other hand, by its quarries, stonemasonry traditions, and buildings (ruins of the Greek Catholic church). The aim of this paper is to present a model for building specialized documentation using a wide range of source materials, methods (field inventory, queries, interviews, high-precision Light Detection and Ranging-LiDAR measurements), tools (Leica ScanStation C10 laser scanner), and techniques (photography, Unmanned Aerial Vehicle-UAV, Terrestrial Laser Scanning-TLS). The applied research procedure model led to the construction of specialized documentation relating to the spatial dimension, natural features, and cultural context of the site. Taking into account the collected data, it should be concluded that the projected geocultural site at Huta Różaniecka, irrespective of the creation of a geopark, has great potential to build a tourist product that is attractive to a wide range of visitors.

**Keywords:** sustainable tourism; geoparks; geoheritage; geotourism; geosite documentation; remote sensing; Roztocze region; Polish–Ukrainian borderland





## 1. Introduction

The process of the creation of geoparks, which started in the 1990s, opened new possibilities of development for areas with unique geological, landscape, ecological, archaeological, or cultural values. The basic tasks of this process were, on the one hand, the protection of geological heritage through the effective protection of geosites, wide promotion of geological sciences, and promotion of their educational and tourist functions and, on the other hand, conflict-free use of values of the area in the local policy of sustainable social and economic development [1,2]. Nowadays, such areas are defined as follows–geoparks are territories with a sustainable development strategy based on the conservation of geoheritage and its use in educational and geotourism activities, together with other natural and cultural resources of the territory ([3], p. 327).

Thus, geotourism is the leading and recommended form of tourism that should be developed in such areas [4]. This form of tourism is understood, in a narrow sense, as "...sustainable tourism with a primary focus on experiencing the earth's geological features in a way that fosters environmental and cultural understanding, appreciation and conservation, and is locally beneficial" ([5], p. 16). In a broad sense, it is defined as "a tourism segment mainly focused on the sustainable usufruct (by geotourists and local communities) of the geoheritage fruition, which can be added to the cultural heritage (material and immaterial) of the areas" [6]. Regardless of the scope of the definition of geotourism, each recognizes the link between geological heritage and its cultural background.

The first geoparks were established as area-based sustainable tourism products in 2004. Currently, in the Global UNESCO Geoparks, there are 195 geoparks in 48 countries, including four transboundary ones [7].

In Poland, such a model of geotourism is not yet common; only two UNESCO geoparks are in operation: the Muskauer Faltenbogen/Łuk Mużakowa, 2009; the Holy Cross Mountains Geopark/Geopark Świętokrzyski, 2021 [7]. In addition, projects of more than a dozen geoparks have been prepared, including–with reference to the analyzed area–Kamienny Las na Roztoczu Geopark [8].

Geosites are the basic form of tourist attractions in geoparks. Their definitions from the beginning of the 21st century describe them as single objects, groups of objects, or geological or geomorphological areas representative for a given region and characterized by outstanding scientific, cultural–historical, aesthetic, and socio-economic values [9,10]. Despite such a broad understanding of the essence of geosites, due to the involvement of geologists and geomorphologists in the creation of geoparks, there have been many studies in the literature in which the scope of the concept has been narrowed only to features related to geology and geomorphology e.g., [11–13]. However, this approach, appropriate for geosites presenting unique tectonic structures, geological phenomena, natural processes, and landforms, has proved incomplete. This is due to the fact that most of the geosites created present the effects of human activity on the natural environment (e.g., quarries, mines). Increasingly, therefore, authors are drawing attention to the need to present geodiversity in the context of its links to the archaeology, history, anthropology and culture of the region, e.g., [14–21]. This has resulted in the use of other terms, including 'geoarcheosite' [22], 'archeogeomorphosite' [23], 'archeo-geosites' [24], 'cultural geomorphosites' [25], 'geoarcheo heritage sites' [26], 'geoarcheomorphosite' [27],'geocultural sites' [28], and 'geocultural landscape' [29], in addition to the term 'geosites', which only considers geological and geomorphological features.

The creation of geocultural sites, as envisaged by UNESCO, involves several important activities and stakeholder groups. The first aspect focuses on the collection and dissemination of geoheritage knowledge and conservation, which is the responsibility of scientific and research institutions. The second aspect concerns the socio-economic sustainability of the area, where the development of geotourism and geoproducts stimulates the creation of local businesses and creates new jobs and sources of income improving the living conditions of the local community and strengthening their identification with the region [30]. Local governments, entrepreneurs, and the local community are responsible for the implementation of this activity [2,31].

Some authors emphasize that such tasks should be carried out especially in areas with high geological and cultural diversity but low levels of economic development [32,33] and/or little known [34,35], as in the case of the eastern part of the Roztocze region [36], in which the village of Huta Różaniecka is located.

The preparation of a geosite carries a great responsibility, especially in the context of knowledge building, its interpretation, protection, and positioning in the planning and management of the area. In addition to explanations of geological phenomena and processes, it is necessary to present the archaeological, historical, anthropological, cultural, ecological, and planning context and to document its current status reliably.

For these activities, new tools and techniques, such as digitization, are increasingly being used in addition to the classic tools and techniques used to date. As many authors have shown, e.g., [13,37–41], this is very helpful for the creation of modern documentation, protection, and popularization of such geosites [38]. A comprehensive overview of the existing and emerging technologies for the characterization and interpretation of geosites is provided by Cayla [42], Cayla, Martin [43], and Quesada-Valverde, Quesada-Román [44].

Rapidly developing techniques for non-invasive light surveying are relevant in this group–Light Detection and Ranging (LiDAR), as well as tools for visualizing, extracting, analyzing, and sharing point cloud data and generating three-dimensional models called digital twins [45]. Terrestrial laser scanning is one of the LiDAR techniques (TLS), that

ensure high detail and accuracy of the information acquired [46]. It allows the preparation of (1) a truthful copy of the site and surrounding area in the form of a 3D point cloud (digital twin), (2) panoramic digital imagery, (3) digital terrain and surface models (DTM, DSM) [47].

Spatial models obtained in this way provide a digital reference and record for future generations [48–51], the creation of high-fidelity replicas of objects, and the creation of multimedia sites with virtual tourism in mind [52,53]. They thus fulfil documentary, conservation, and educational tasks. They can also function undue as virtual tourist attractions [54,55].

The main objective of this paper is, with reference to UNESCO's objectives, to present an example of collecting and building geocultural site expertise, with Huta Różaniecka as an example. The paper will present various methods, tools, and techniques as well as data sources for building the most complete geocultural site context. We assume that the presented model of building specialized documentation and systematized knowledge can be used by local governments to prepare geotourism products and, consequently, to develop geotourism. The knowledge offered may encourage cooperation between scientific institutions and local/regional stakeholders responsible for the creation of geoparks. Such collaboration, within the framework of sustainable development, is suggested by UNESCO for areas of natural and cultural value–geoparks and/or international biosphere reserves [56]. The specific objectives, in addition to evoking traditional methods of documenting geosites, focus on pointing out the usefulness of modern tools and techniques in collecting and building the knowledge about geocultural sites.

## 2. Study Area

Huta Różaniecka is a small village located in the Podkarpackie Voivodeship in the eastern part of the Roztocze region. It is situated in the vicinity of the Poland–Ukraine border (Figure 1).

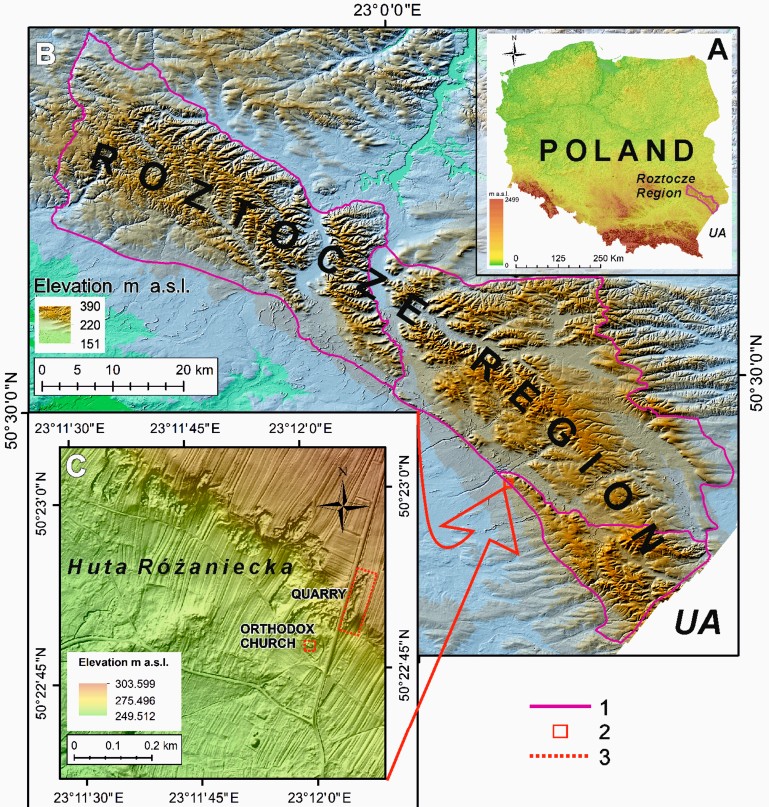

**Figure 1.** Location of the study area: (**A**) location of the Roztocze region in SE Poland; (**B**) location of the Roztocze region: 1. boundaries of the physical—geographic region according to Solon et al. [57];

(**C**) Overview of the study area in Huta Różaniecka village: 2. the study area against the background of the Roztocze region, 3. the geocultural sites analyzed; UA—Ukraine.

In regard to relief (topographic) features, this is the highest elevated part of the Roztocze region. The altitudes reach 390 m above sea level (m a.s.l.), with average relative heights of 96 m. Deep gorge valleys dissecting its western edge are also characteristic. The isolated hills of Wielki Dział, Długi Goraj, and Krągły Goraj are above the highest surface level (350–360 m a.s.l.) [58]. There are relatively many small single tors (e.g., Diabelskie Kamienie near Werchrata), caves (e.g., Przyjaciół), and epikarst pipes [59]. This part of the Roztocze region is formed by Miocene detrital limestones, shellstones, sandstones, and sands [60–62]. Quaternary sediments fill the river valleys. Locally, there are loess covers and sand dunes [63]. The eastern part of the Roztocze region is rich in groundwater and surface water. It is the headwater area of the Tanew and Rata Rivers [64]. The landscape is dominated by forests, and arable land prevails on the highlands. River valleys and depressions are used as meadows and pastures [65].

Due to its border location, the eastern part of the Roztocze has been subject to various influences over the centuries, most often of a military nature. Historical conditions meant that the border between Poland and Kievan Rus', Russia and the Austro-Hungarian Empire, the German Reich and the Soviet Union, Poland and the USSR, Poland and the Ukraine ran through the area (Figure 1A,B). Today, the region comprises the eastern border of the European Union. The affiliation to various state organisms, which began in the early Middle Ages, was decisive in shaping the multinational and multicultural character of the area. Until the 1940s, Poles, Ruthenians/Ukrainians, Jews, and Germans lived here [66]. The irreversible change in the ethnic and confessional structure, which is nowadays almost homogeneous, occurred as a result of the events related to World War II and the political decisions taken at that time [67].

Despite the destruction, objects of multicultural heritage have survived in the region, which are currently increasingly being used for the development of various forms of tourism. An example of such activities is the theme undertaken in this article, concerning the village of Huta Różaniecka (Figure 1C).

## 3. Materials and Methods

The projected geocultural site at Huta Różaniecka includes quarries of Miocene detrital limestones, sandstones, and bivalve casts and shells and the ruins of the 19th century St. Nicholas Greek Catholic Church built from these rocks. They were selected from among 166 geosites prepared for the Kamienny Las na Roztoczu Geopark. It is planned to include 81 geological and geomorphological sites (e.g., petrified forest sites, individual hills, rocks, caves, springs, valleys, gorges, gullies), 55 cultural sites (e.g., quarries of Miocene rocks, exposures of Quaternary loess and sands in dunes related to their exploitation), and 18 geocultural sites (e.g., cemeteries with tombstones, secular and sacred buildings made of local rocks in local quarrying centres) [8]. The uniqueness of the projected site at Huta Różaniecka is due to the fact that it is the only one among the mentioned sites to combine former rock extraction sites and a building made from these rocks.

In creating the geocultural site at Huta Różaniecka, reference was made to the recommendations of the method of developing geosites adopted at the end of the 20th century by ProGEO (The European Association for the Conservation of the Geological Heritage). These featured making an inventory and documentation including the name and type of the site; location; features in the context of the geological region; a concise description of the main elements with basic literature; drawings; photographs; type and chronostratigraphy; comparison and justification of value in relation to other sites; information on conservation status; threats; information on significance for the history of geological research; landscape; culture; and education [9,68].

During the work on the geocultural site project, a wide range of data sources were used and acquired using a variety of methods, tools, and techniques. In the first stage, the preparation of the inventory card, the focus was placed on acquiring knowledge from secondary sources: archival/historical (e.g., plans, maps, photographs/engravings of sites, compilations of statistical data, regional studies) and contemporary (e.g., geological maps, topographic maps, numerical elevation model (DEM), data geoportals, literature). By means of a search method, information was collected on the features of the geological structure, the history of the village and the characteristics of its inhabitants, the economic functions of the village, including the tradition of mining local rocks, and the legal status of the property (Figure 2).

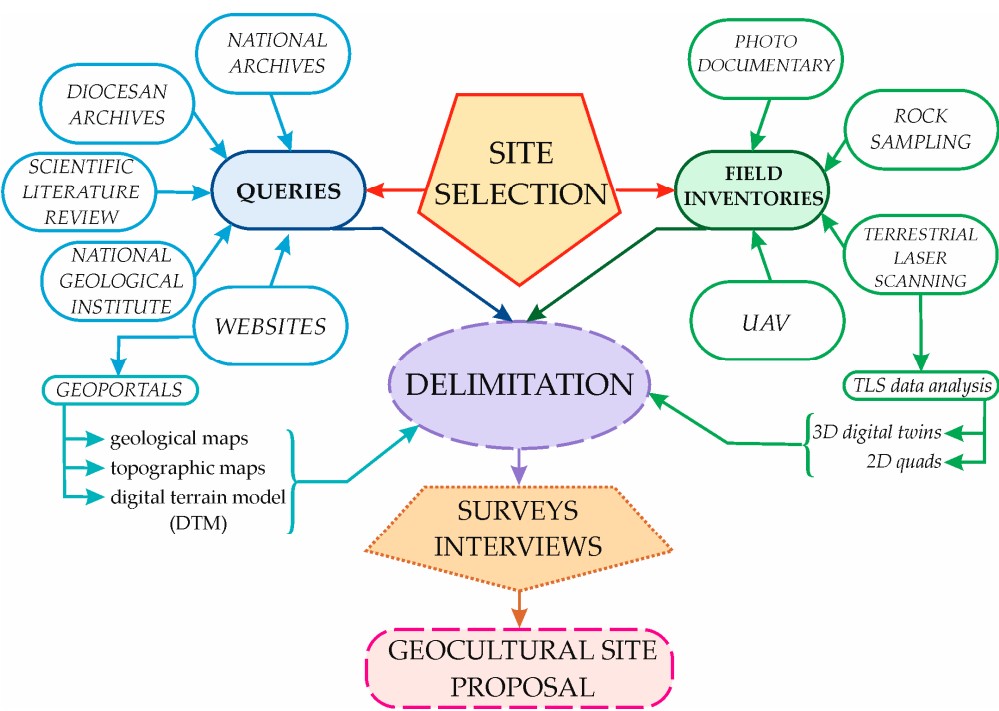

**Figure 2.** Diagram outlining the research process used to collect data.

Other missing content required the acquisition of primary materials. The field inventory made it possible to determine the number of rock exposures, the location of quarries and church (geographical coordinates), their accessibility, their state of preservation, their location in relation to existing tourist routes and trails, the existing threats, and the possibilities of their development for geotourism. In addition, the characteristics of the rocks found in the disused quarries were verified against their descriptions in the literature. Macroscopically, the features of the rock material used for the construction of the church were compared with the features of the rocks exposed in the neighboring quarry (Figure 3).

In parallel with the inventory, eight local government representatives were interviewed. They represented the Poviat Starosty in Lubaczów (4 persons), the Town and Commune Office of Narol (3 persons), and the Geotourist Association "Roztocze Wschodnie" (1 person). A standardized set of 10 questions was prepared for the respondents: Have you heard about the Kamienny Las na Roztocze Geopark project? What resources are responsible for the potential of the geopark? Can you indicate strategic documents addressing the geopark project? Have you heard about the planned geosite in Huta Różaniecka; if so, what assets does it have? Can elements of this geosite be used for the development of tourism/geotourism; if so, why? How do you think the development of tourism/geotourism can influence the local community? If the geopark were to be created, how would you prepare the space of this geosite? What scope of activities should be undertaken to create a tourist product based on this geosite? Are there any risks associated

with the activities aimed at creation of the product; If so, what difficulties do you see? How would you provide financing for this project? The responses showed the respondents' opinion about the possibility of creation of a geotourism product in Huta Różaniecka. The last stage of the fieldwork included photographic documentation and high-accuracy terrestrial laser scanning surveys (TLS) (Figures 4 and 5).

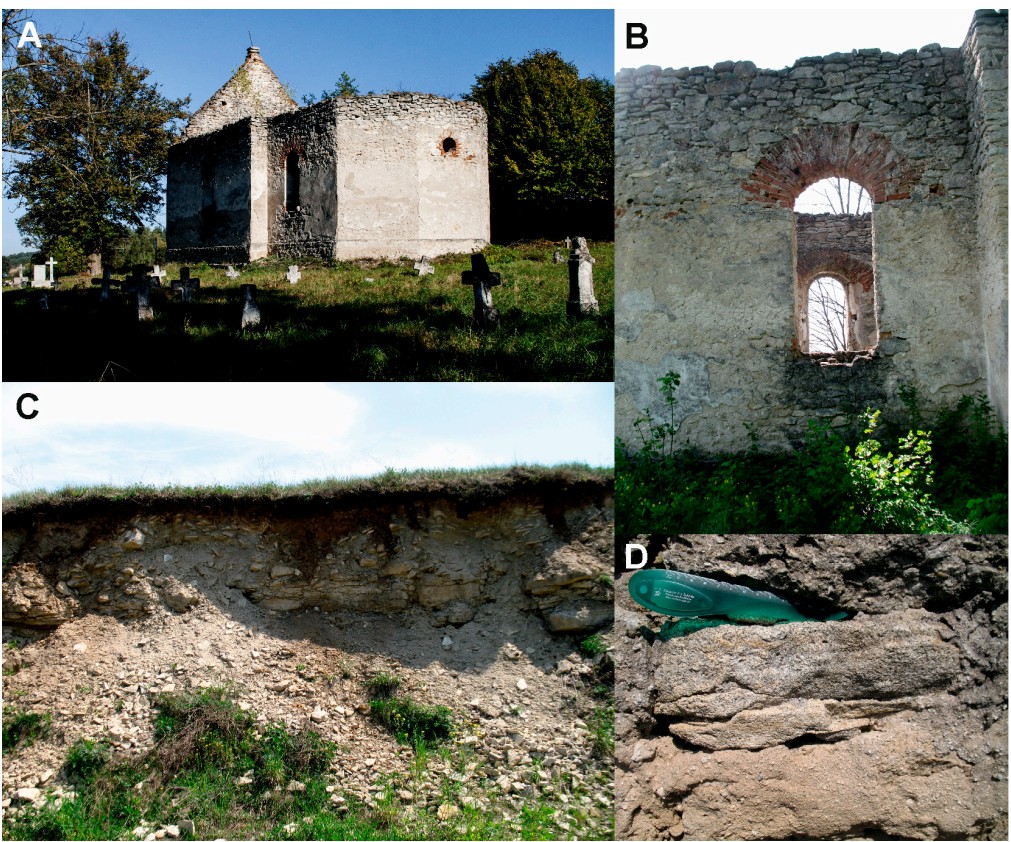

**Figure 3.** (**A**,**B**) St. Nicholas Greek Catholic Church in Huta Różaniecka; (**C**) quarry of Miocene shellstones and detrital limestones, (**D**) features of rock built into the church wall in Huta Różaniecka.

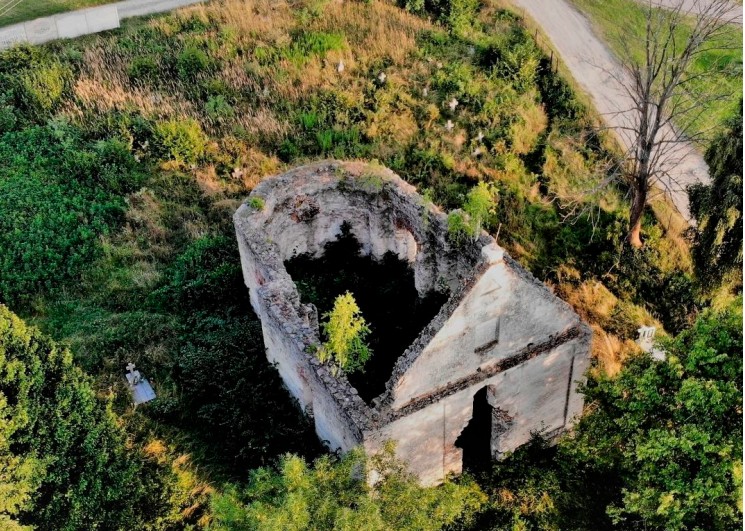

**Figure 4.** Contemporary state of the St. Nicholas Greek Catholic Church and the cemetery in Huta Różaniecka, phot. Marcin Mazurek.

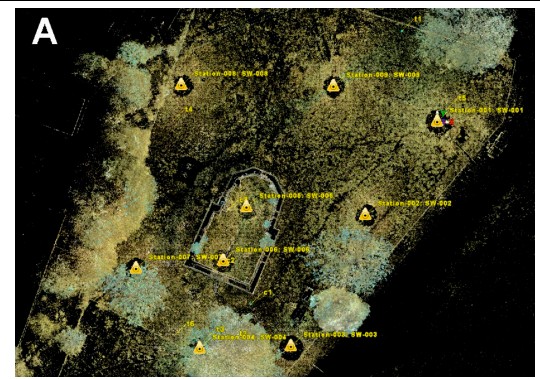
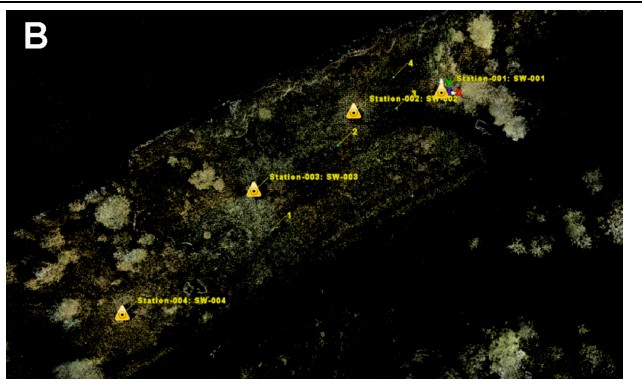

**Figure 5.** Overview of studied geocultural sites. Yellow triangles mark the location of the Leica ScanStation C10 terrestrial scanner station: (**A**) St. Nicholas Greek Catholic Church in Huta Różaniecka; (**B**) Huta Różaniecka quarry.

TLS surveys of the building and quarry were carried out using a Leica ScanStation C10 terrestrial laser scanner (Figure 5A,B). The measurement accuracy was 4 mm (up to 50 m) for the distance and 6 mm for the 3D location [69].

The scanning of the St Nicholas Greek Catholic Church and its surroundings was carried out from nine sites (Figure 5A). The nearby quarry was scanned from four sites (using 4 TPs) located along its edge (Figure 5B). The data were processed in three stages (registration → noise removal and filtering → 3D model extraction) using Leica Cyclone 8.1 software. As a result, two digital twins of the studied objects were produced: (1) a 3D model–the church with its surroundings and (2) a 3D model of the quarry.

The collected materials were collated and arranged in the course of field works. As a result, almost complete geological and cultural characteristics of the projected site were obtained, the geological map of the Huta Różaniecka surroundings was supplemented and updated, and a virtual model of elements of the geosite site was prepared. The conditions and possibilities of development of geotourism on the basis of the designed site were analyzed.

## 4. Results

The results of the desk research and field studies obtained showed the scientific, educational, and applied character of the planned geocultural site. It turned out to be interesting in terms of content related to the geological history of the area. It is also a clear example of the centuries-old use of local rocks in human economic activities.

Another aspect was the verification of the application of modern tools and techniques for building of knowledge, documenting sites, and assessing the possibility of creating a tourist product.

### 4.1. Geological Features

The planned site at Huta Różaniecka is interesting for two reasons. The first is the geological history of the area. Its record is contemporarily contained in the exposure of the tectonic and erosional edge separating the Roztocze region from the Sandomierska Basin region [70]. In the Miocene, it was located in the northeastern shallow shelf zone of the Tethys Ocean. The edge, together with the adjacent Roztocze area, was uplifted in the declining phase of the Alpine orogenesis. It is therefore one of the elements of the inversion relief of the region and is currently well visible in the landscape (Figure 6). Within the limits of the planned site, it reaches a height of a few to a maximum of 30 m and a length of about 12 km, extending on both sides of the road that crosses the village (Figure 7). More than a dozen rock exposures, former mining sites (Figure 1C) with varying state of preservation and accessibility, have been located along this segment of the edge.

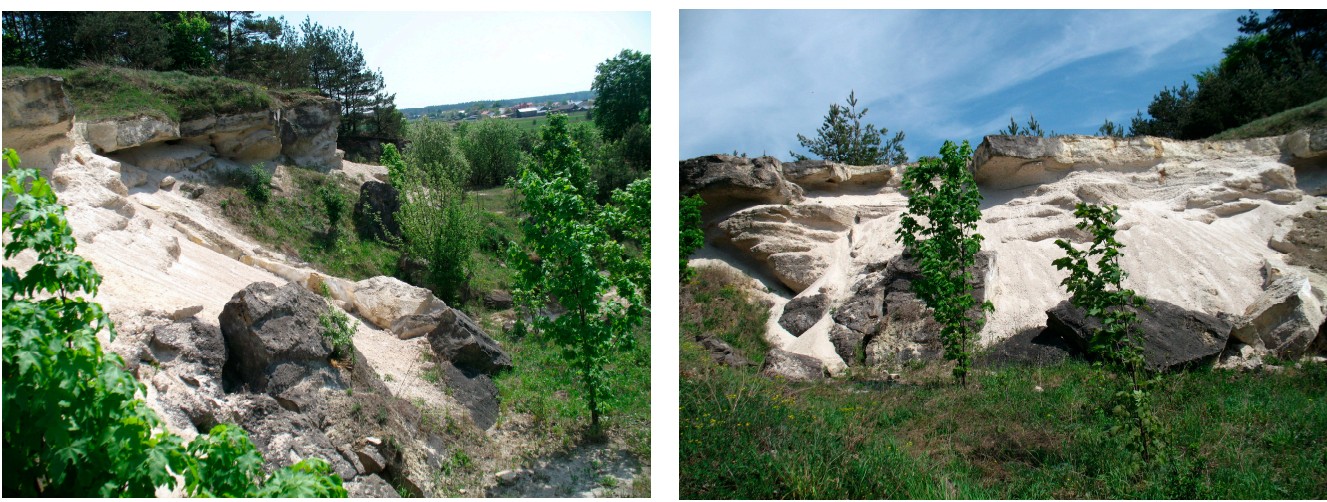

**Figure 6.** View of the tectonic-erosional edge separating the Roztocze and Sandomierska Basin regions.

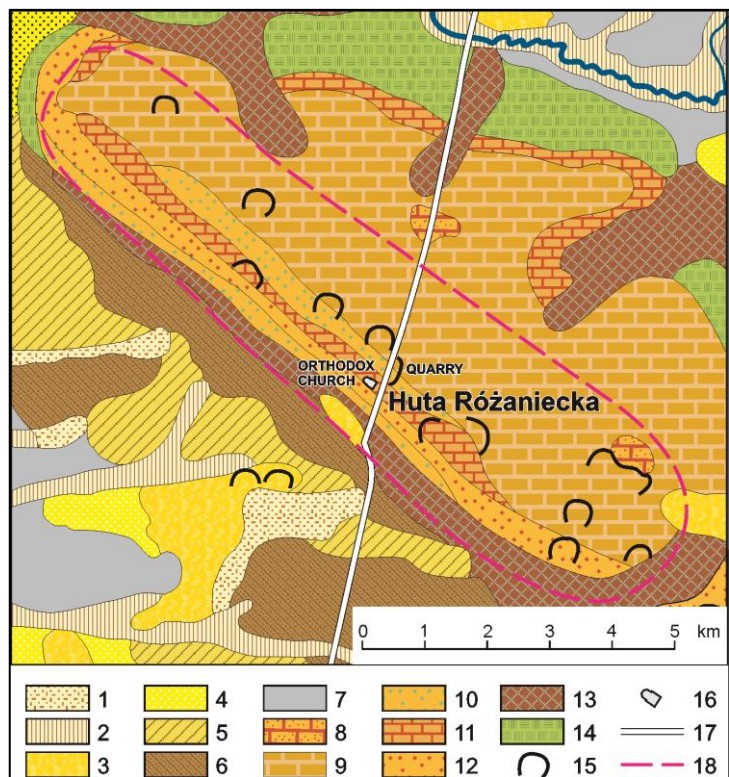

**Figure 7.** Geological structure of Huta Różaniecka surroundings (after [62,71], supplemented): Quaternary: 1–humic sands in valleys and hollows without outflow, 2–river sands of flood terraces 3.0–5.0 m a.l. river, 3–aeolian sands in the dunes, 4–aeolian sands, 5–aeolian sands on the tills, 6–deluvial sands on the tills, 7–river sands in non-flooded terraces 5.0–10.0 m a.l. river; Miocene: 8–limy fine-grained sandstones, 9–detrital limestones, 10–limestone sands with detritus, 11–algae-shells limestones (*Lithothamnium*) with rhodolithes, 12–quartz sands, 13–glauconite sands; Campanian: 14–siliceous rocks (gaizes); 15–quarries; 16–location of the church ruins; 17–road; 18–geosite boundaries.

Here, rocks and sediments whose features document the palaeoenvironment of the shallow Miocene sea are spectacularly exposed, as illustrated by the revised contents of the geological map (Figure 7).

In the planned site, the phases of sedimentation at that time can be easily and pictorially reconstructed on the basis of the features of the rocks and sediments of the Miocene sea. In

the upper part of the geosite, bivalve casts and shells (coquina bed) with an accumulation of bivalve casts (*Glycymeris* sp. *Chlamys* sp., *Neopycnodonte* sp.; [72]) and scallop and oyster shells are visible. They are recognizable, conducive to the Miocene, and easy to find. Below, quartz-rodolithic sediments occur [73]. The rhodoliths they contain (4–8 cm in diameter) are the remains of a colony of spherically shaped kelp that grew on the seabed at that time. n the conditions of rapid, turbulent water flow, they were detached from the seabed and transported into the shallow bay. The quartz-rodolithic [sediments are overlain by strongly cemented sandstones (Figures 7 and 8). Black flint and phosphorite pebbles are occasionally present in these formations [62]. In the exposure, the bedded part of the quartz-rodolithic sediments is not visible. There are also currently no sources to interpret them.

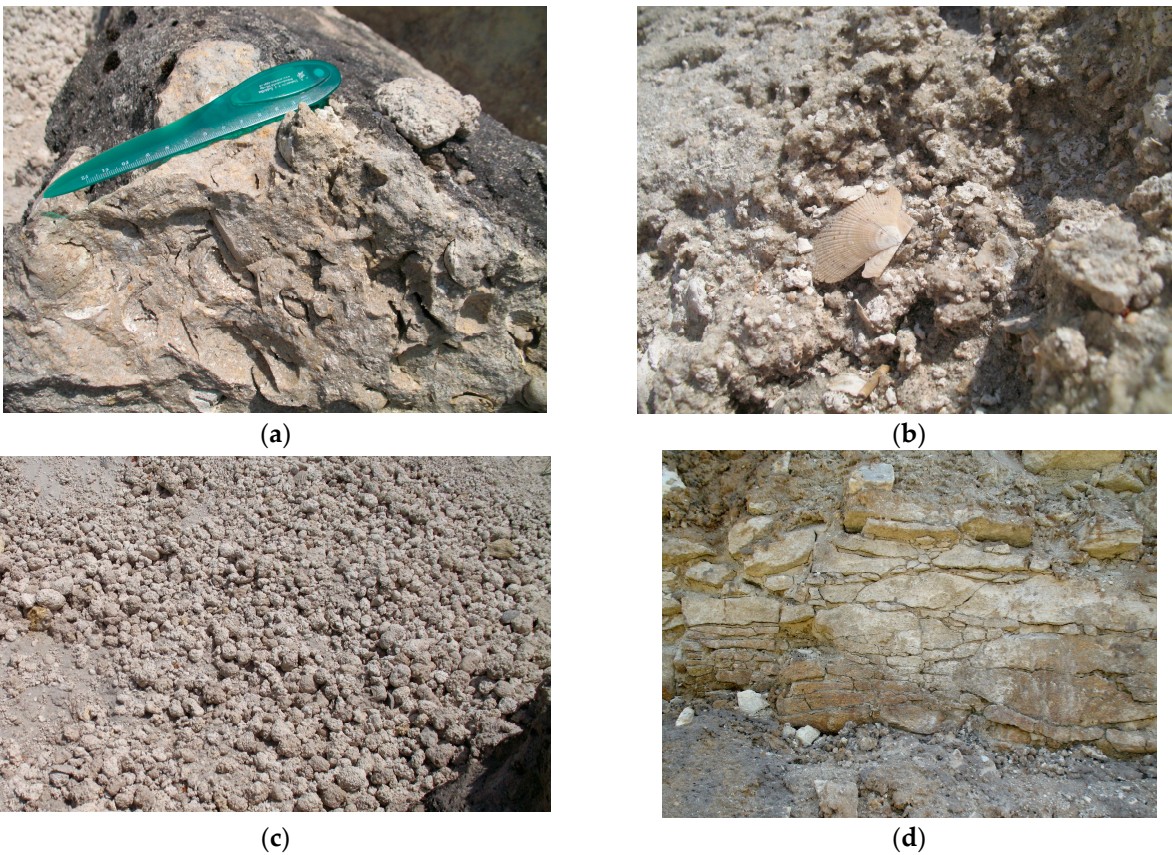

(**a**)　　　　　　　　　　　　　　　　(**b**)

(**c**)　　　　　　　　　　　　　　　　(**d**)

**Figure 8.** Miocene rock types at the Huta Różaniecka quarry: (**a**) shellstone; (**b**) scallop in quartz-rodolithic sediment; (**c**) rhodoliths; (**d**) limy sandstone.

In the immediate vicinity of the site, organodetrital limestones are exposed (Figure 7), in the ceiling of which, in the weathering, pockets of karst–epikarst can be observed [74].

Another important aspect of the site is the centuries-long use of the rocks in the local economy. The raw material, lying shallow beneath the ground surface, was easy to extract, and its physical characteristics determined its various uses, e.g., in construction (foundations, fences, buildings) and for paving roads. The exploitation intensified in the mid-19th century, when peasants were enfranchised in the Austrian partition. As a result, rock extraction sites were located within the boundaries of the land belonging to the peasants, which made the raw material cheap and easily accessible.

Miocene organodetritic limestone and shellstone and calcareous sandstone were particularly suitable for construction. The exploitation was facilitated by their blockiness associated with lithological variation and a dense network of vertical rock fractures/joints (Figure 8d).

The physical properties of the limestones and limy sandstones were important for the use thereof for construction. Limestones are made of organic structures contributing to the dominance of $CaCO_3$ in the rock (90–98%). The presence of algal bands separated by bioclasts has an impact on the high separateness of rock layers [73]. Since the pores are filled with a calcite binder, some layers are relatively massive (average porosity of 17.4%). This contributes to low weight absorbability (4.85%) and volumetric absorbability (10.85%) as well as high compressive strength (90.95 MPa in the dry state, 60.09 MPa after water saturation) [75].

The different-grained limy sandstones are characterized by variable contents of $CaCO_3$ (38–64%), $SiO_2$ (30–50%), and CaO (20–30%). Nevertheless, they are massive (with a porosity value of 5.99%). They exhibit low absorbability (weight absorbability–1.28%; volumetric absorbability–3.22%) and high compressive strength–80.22 MPa in the dry state and 60.52 MPa after water saturation [75].

The features of this exposure, both those relating to the history of the Earth and those related to human activity, are of very high scientific and educational value. They therefore represent an excellent potential for the construction of a geocultural site.

### 4.2. Cultural Features

The local exposures are connected with the history of the village of Huta Różaniecka. The area of eastern Roztocze, where the village is located, has a long history of settlement. The oldest traces of human presence date back to the Late Paleolithic. In the Mesolithic, the area was inhabited by hunters from the Janisławice culture group. At the beginning of the Neolithic period, the area was occupied by representatives of the Lublin-Volhynia agricultural painted ceramics culture. Next, the region was inhabited by shepherds and farmers of the Funnel Beaker, Globular Amphora, and Corded Ware cultures. The area was already largely inhabited in the Early Bronze Age (1450–1200 BC), as evidenced by the artifacts of the Trzciniec culture. Representatives of the Lusatian culture appeared here in the middle of this era (1200–1000 BC). In the following centuries, the region was sparsely populated. The settlement network became denser only from the beginning of the Early Middle Ages (6th century). Settlement intensified especially before the mid-10th century. The settlement network of that time was destroyed as a result of numerous invasions in the 13th century [76]. Its reconstruction began in the following century, with the greatest expansion in the 16th century. Wars and invasions that took place in the 17th century again slowed down the pace of settlement development [77]. This changed after the end of armed conflicts in the following century, i.e., a period when Huta Różaniecka was established.

The land on which the modern village is located was an area with difficult features for settlement. Poor communication accessibility resulting from the dense forest complexes of the Puszcza Solska and Roztocze and the poor soils were a great challenge for the future inhabitants from the very beginning. For this reason, the attempts at settlement were closely linked to the industrial exploitation of the forests. Coal mills, tar mills, shingle mills, ore mills, and glass factories were organized on the basis of local forest resources, consuming huge amounts of wood. Around them, settlements developed over time, whose inhabitants burned ashes, made potash, melted tar, burned coals, dug bog ore, smelted iron or glass, and practiced wood crafts–shingling, wheelwrighting, cooperage [78]. Such were the beginnings of the described village.

It is known that, before 1621, unskilled laborers settled in the area and were given concessions/free labor for a year in return for clearing the forest for buildings and gardens [78]. In 1713, a small glassworks was established here, after which the resulting settlement was called Huta Płazowska, Huta Prosta Płazowska, Huta Różaniecka, or Huta Lubliniecka [79]. In 1729, there were 25 settlers' houses in the settlement established on cleared glades. It was the beginning of a new royal village within the boundaries of the Lubaczów starosty. It was distinguished by a characteristic spatial layout, referring to the economic activity of its inhabitants. The village consisted of a central part, with a chain layout, and several hamlets: Banachy, Huta Stara, Korkosze, Maziarnia, Podsigły, and Rebizanty [80].

The glassworks operated until the 1850s, and then ceased operations. It resumed operations in the early 1880s as Huta Różaniecka [81], p. 27.

In 1818, the village of Huta Różaniecka was sold to the family of Count Brunicki and from then on it became part of the Ruda Różaniecka estate. At the beginning of the 20th century, it passed into the hands of the Wattmann barons Maelcamp de Beaulieu [80].

Thus, the village had an agricultural and industrial character from the beginning. It functioned on the basis of the resources of the Solska Forest in the Sandomierz Basin–forests and glass sands–and of the Roztocze region–forests and rocks (Figures 1 and 7).

The declining forest areas as a result of centuries of exploitation contributed to the development of agriculture. This influenced the structure of land use. In the 19th century, arable land predominated within the village boundaries (Figure 9). In addition, from the 19th century onwards, Huta Różaniecka became a site for the exploitation of local rocks. The access to the rocks became easier after the enfranchisement of the peasants in the second half of the 19th century. The resources were then located within the boundaries of their fields–in the northern Roztocze part of the village (Figure 10). From then on, mining and processing of the rocks became an additional source of income for the inhabitants. This economic activity continued until the end of the 20th century [78].

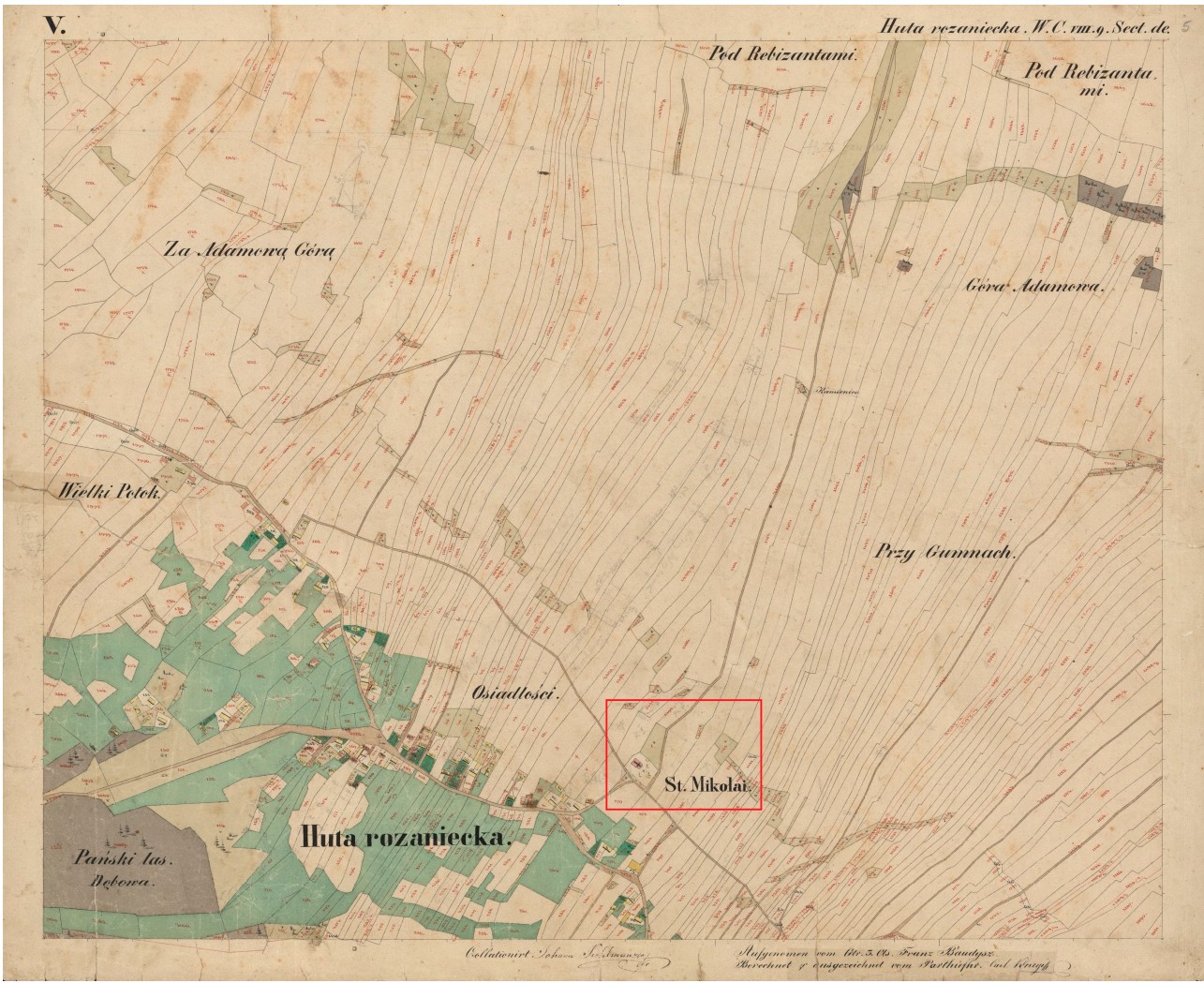

**Figure 9.** Archive plan of Huta Różaniecka village in 1854 [82]. The red color indicates the location of the church and the adjacent quarries.



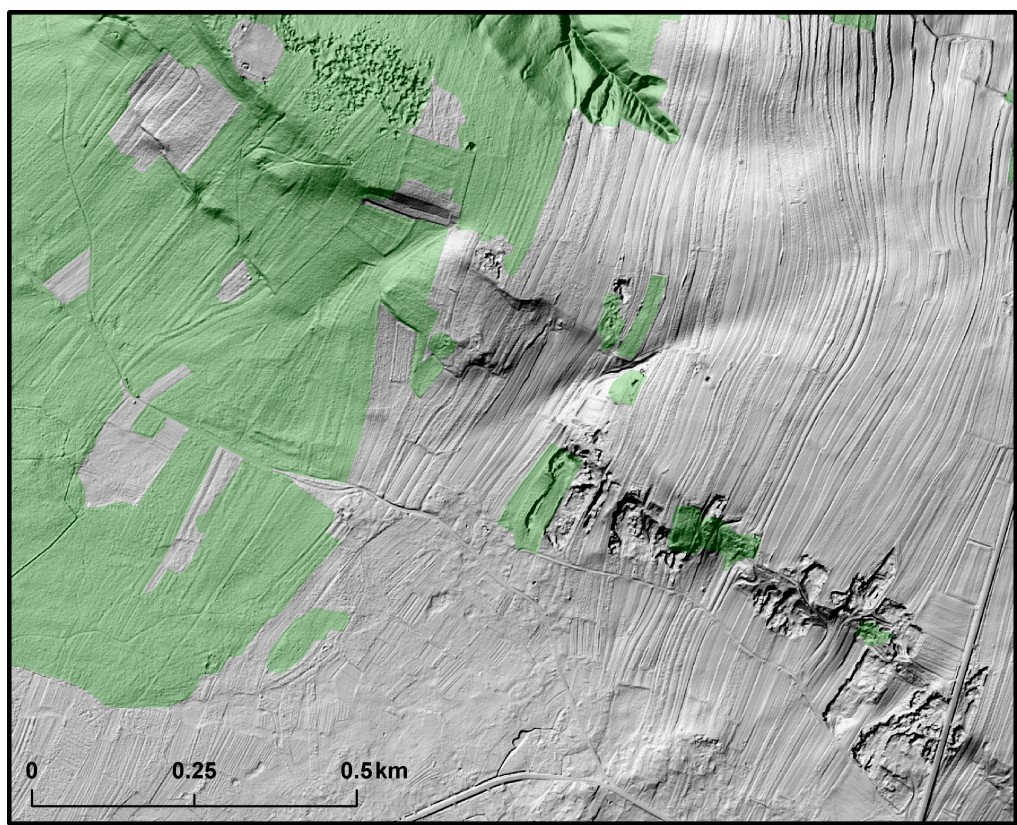

**Figure 10.** Examples of non-existent contemporary local rock exploitation sites visible on LiDAR. Forests are shown in green. Square-grid DEM data with five-meter resolution was obtained from the publicly available resources of the Head Office of Geodesy and Cartography (GUGiK). Forest areas were obtained from the Database of Topographic Features (BDOT10k). The content and detail of the BDOT10k database generally corresponds to a traditional topographic map at a scale of 1:10,000. The source dataset was referenced in the PL-1992 system (National Geodetic Coordinate System 1992 for Poland).

In the context of the proposed geostationary site, in addition to the tradition of rock mining, the heritage associated with the socio-cultural features of the villagers, i.e., the ruins of the St Nicholas Orthodox Church (Figure 3A,B and Figure 4), is important.

Before World War II, Huta Różaniecka was inhabited by about 1000 people. It is known that in 1921 it had a population of 1250, of which 1234 were Polish and 16 were Ruthenians. In terms of the religious structure, there were 954 Roman Catholics, 288 Greek Catholics, and 8 Jews [83]. In 1939, the village already had 1400 inhabitants, including 1195 Poles, 200 Ukrainians, and 5 Jews [84]. Thus, the most numerous group were Poles (Catholics), while the others, Ukrainians (Greek Catholics) and Jews, were less numerous.

Until 1926, there was no Roman Catholic church in Huta, and Catholics attended the church in Plazov. However, the Greek Catholics, built a wooden church in the village as late as 1728 [85]. Another one was erected on a new site from the foundation of Rev. Leon Lebedynski. It was a small temple founded on a tripartite plan. By the end of the 18th century, it was in very poor condition and was demolished after 1825 [80]. In its place, with the help of the village owner Jan Baron Brunicki, a brick church was erected from local rocks in 1835 (Figure 11). It served a small group of Greek Catholic worshippers also from the surrounding villages and hamlets of Rebizanty and Korkosze [86,87]. The St Nicholas Greek Catholic Church was destroyed on 26 June 1943 as a result of a German pacification operation [87].

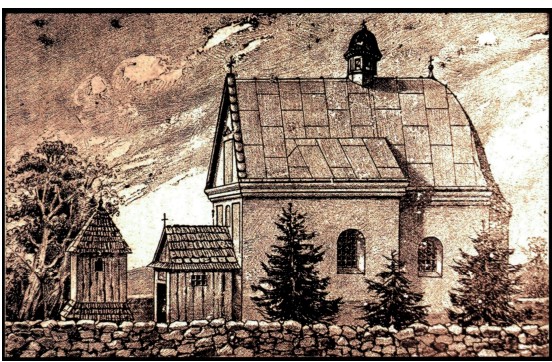

**Figure 11.** The appearance of the St. Nicholas Greek Catholic Church in Huta Różaniecka in 1896 based on a replica of a print made by Marcin Materniak. The original is kept in the collection of Vasyl Svobodian.

World War II irreversibly changed the size of the village population and its ethno-religious structure. Between 1943 and 1944, the village population suffered significantly as a result of German repression. The Germans deported Poles and Jews to concentration camps and deep into Germany for forced labor. After the end of the war, the Ukrainian population was also displaced as part of the Operation Vistula (1945–1947) [88].

The result of the village's turbulent history is a drastic reduction in its population to 323 people in 2021 [89], i.e., only about 23% of the 1939 population. They are homogeneous in terms of nationality (Poles) and religion (Catholics). A reminder of the former multi-cultural community are the ruins of the Greek Catholic church and the gravestones in the surrounding cemetery (Figure 3A,B and Figure 4). These objects attract the attention of tourists. They are located along busy traffic and tourist routes. They belong to a group of very few sacred buildings built from local rocks in Roztocze.

*4.3. Attitudes of Local Authorities towards the Planned Geostationary Site*

The interviews conducted with local government representatives resulted in the following observations. The local authorities have varying degrees of knowledge of the natural value of the existing exposures. They have a general knowledge of the geological and historical past of the area. They perceive the uniqueness of the planned geosite. They link it to the Kamienny Las na Roztoczu Geopark project. They are able to point to strategic documents in which this project is included. They treat the preserved ruins of the Greek Catholic church and rock mining sites as their heritage. The result of this attitude is a desire to promote this place and build on its potential as a tourist attraction.

The representatives of local authorities and associations note the possibilities of using a potential tourist product in the socio-economic development of the village/area. In their view, it could stimulate the establishment of local businesses and create new jobs and sources of income, thereby improving the living conditions of the inhabitants and strengthening their ties with the region.

They see a potential geocultural site as a landscaped space including prepared and pro-tected rock exposures with plaques (with specialized geological explanations/descriptions), fossil extraction sites, thematic paths linking the sites where different rocks have been extracted, and thematic plaques introducing the history of the village and describing the ruins of the Greek Catholic church.

They are aware of the wide range of activities needed to prepare this tourist attraction, including spatial, legal, ownership, scientific, technical, promotional, and economic aspects. Nevertheless, they are determined to undertake them.

They also recognize the many difficulties involved, e.g., the problem of acquiring and securing adequate funding, unregulated land ownership issues, ensuring the safety of tourists in the exposures, the problem of acquiring adequate information about the natural and cultural features of the site, the concerns about cooperation with the scientific

community, the need to secure the geosite in relation to environmental principles, the attitude of local residents towards tourism, and the poor recognition of the area.

The success of the geocultural site is closely linked to the success of the planned geopark, which would provide the necessary funding.

### 4.4. Application of TLS for Acquisition, Processing, and Visualization of Data

For the St. Nicholas Greek Catholic Church ruin and the quarry in Huta Różaniecka, the Digital Terrain Model (DTM) was developed applying the survey procedure described below and based on the acquired point cloud data. The measurement data obtained in the 'point cloud' form and imported to the Cyclone 8.1 program formed the 'model space'. To obtain a full three-dimensional model of the scanned area, the individual 'model spaces' were registered into a unified 'Scan-World' using the TP procedure [90]. From 3 to 7 target points (TPs) were used for registration of the individual scans, which ensured high accuracy. For the surroundings of the church, the mean horizontal error usually did not exceed 2 mm, the vertical error was lower than 0.1 mm, and the mean absolute error MAE was 2.1 mm. The values for the quarry were 2.6, 0.01, and 2.7 mm, respectively (Table 1).

**Table 1.** Characteristics of point clouds and range of errors.

| St. Nicholas Greek Catholic Church in Huta Różaniecka | | | | | |
|---|---|---|---|---|---|
| | **No. of Coincident** | | **Errors** | | **Point Cloud** |
| Scan Name | TP [1] | SW [2] | Horz [3] [mm] | ver t [4] [mm] | MAE [5] [mm] | No. of pts [6] |
| SW1 | 5 | 8 | 1.86 | −0.01 | 1.98 | 5,702,080 |
| SW2 | 6 | 7 | 1.91 | 0.03 | 1.96 | 6,551,921 |
| SW3 | 7 | 6 | 1.86 | −0.07 | 2.02 | 7,840,854 |
| SW4 | 4 | 4 | 2.30 | −0.13 | 2.31 | 10,023,425 |
| SW5 | 3 | 2 | 2.40 | −0.23 | 2.48 | 9,506,355 |
| SW6 | 3 | 2 | 1.30 | −0.35 | 1.38 | 10,008,801 |
| SW7 | 3 | 2 | 5.23 | −0.33 | 5.27 | 9,473,787 |
| SW8 | 3 | 1 | 2.03 | −0.10 | 2.03 | 6,713,916 |
| SW9 | 3 | 7 | 2.61 | 0.02 | 2.63 | 5,911,665 |
| Mean | | | 2.05 | −0.07 | 2.13 | * 71,732,804 |
| Quarry in Huta Różaniecka | | | | | | |
| SW1 | 4 | 3 | 2.80 | 0.02 | 2.87 | 5,624,328 |
| SW2 | 4 | 2 | 2.80 | 0.01 | 3.00 | 5,696,662 |
| SW3 | 3 | 1 | 1.10 | −0.17 | 1.20 | 5,350,677 |
| SW4 | 3 | 3 | 2.02 | 0.00 | 2.22 | 5,176,771 |
| Mean | | | 2.56 | −0.01 | 2.68 | * 21,848,438 |

[1] Target Point; [2] ScanWorld; [3] Horizontal; [4] Vertical; [5] Mean Absolute Error; [6] number of points; * Total number of points.

After registration of all point clouds, high-resolution three-dimensional (3D) digital twins (DTs) were obtained (Figure 12A,B).

The 3D model for the church surroundings was generated by registration of 9 scans containing from 5.7 to 10 million points (Mpts), which yielded a 3D DT consisting of over 71.7 Mpts after registration. The 3D DT for the quarry was generated from 4 scans (from 5.1 to 5.7 Mpt) registered into a 3D model consisting of 21.8 Mpt (Table 1). This large number of points facilitated the analysis of the parameters of the studied objects (e.g., detailed 2D and 3D measurements), extraction of 3D DT for the ruins and quarry walls (Figure 12), and making lateral and vertical quads of the church ruins (Figures 13 and 14).

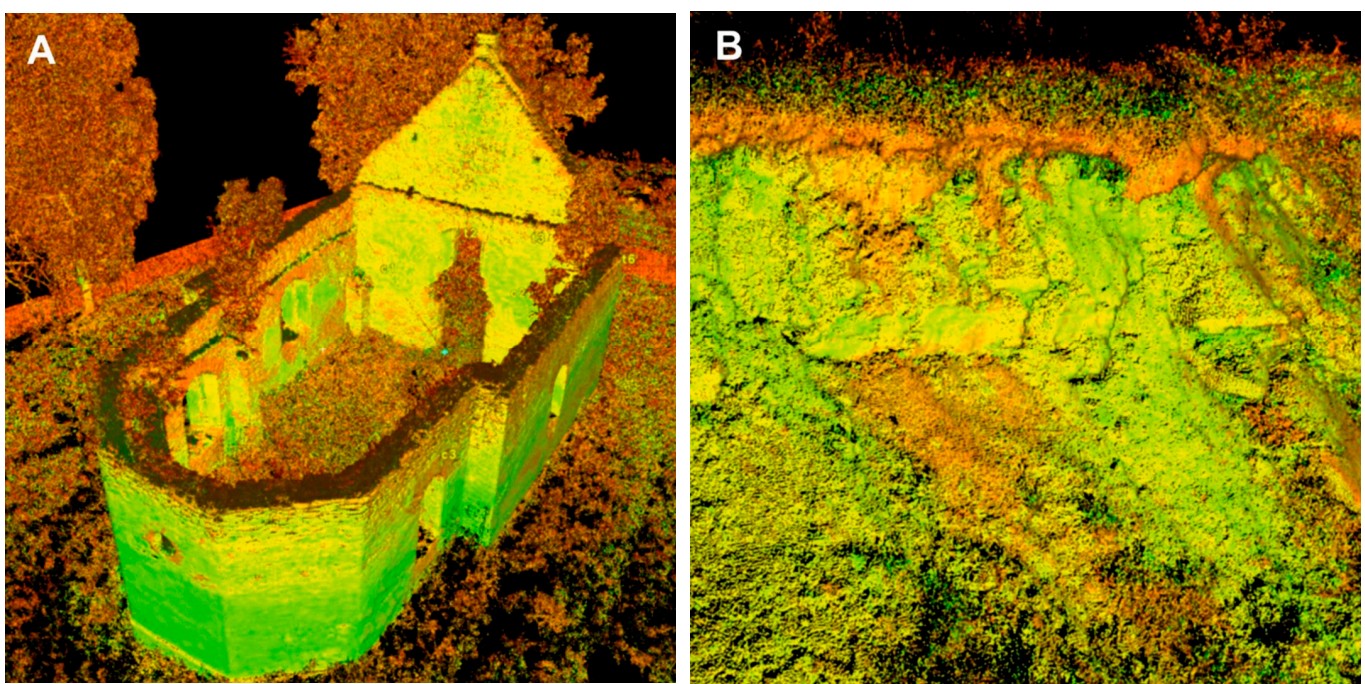

**Figure 12.** High density three-dimensional (3D) digital twins (DTs): (**A**) St. Nicholas Greek Catholic Church, (**B**) epicarst forms in quarry.

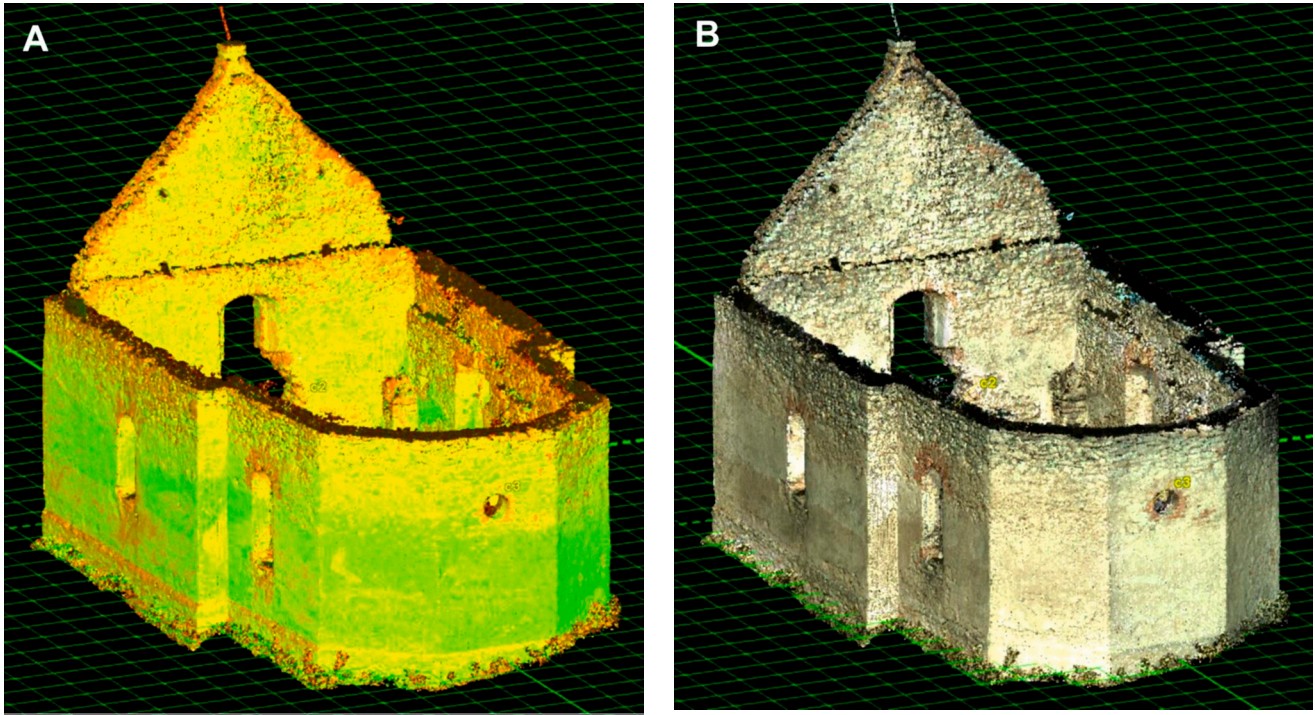

**Figure 13.** Three-dimensional (3D) digital twin (DT) of St. Nicholas Greek Catholic Church: (**A**) in reflective intensity colors; (**B**) in natural colors (RGB).

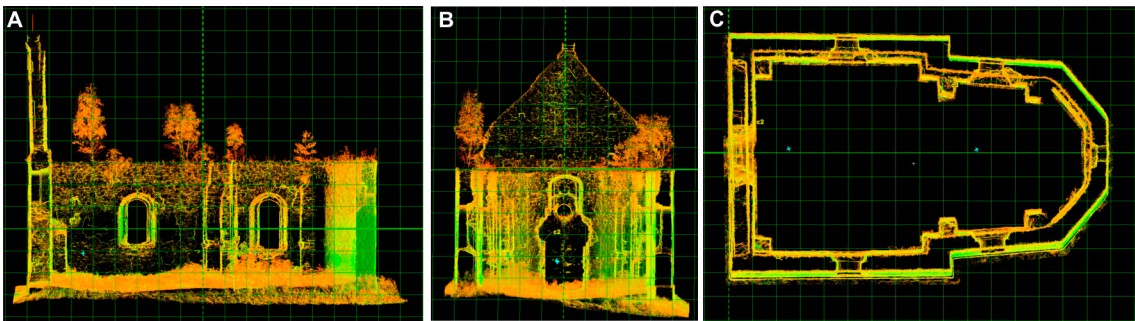

**Figure 14.** Examples of quads of the St. Nicholas Greek Catholic Church: (**A**) lateral view, (**B**) front view, (**C**) top view. Grid lines every 1 m.

## 5. Discussion

The studies show that areas with unique geological-paleontological, cultural, archaeological, historical, and ecological values, including those economically underdeveloped and little known, may have their chance for development through, e.g., geotourism, as recognized and recommended by Eder, Patzak [1], Alexandrowicz [2], Azman et al. [91], Fassoulas, Zouros [92], Farsani et al. [32], Wójtowicz et al. [93], Ólafsdóttir, Dowling [94], and Kubalíková [34]. In this trend, however, the most important thing is to perceive the geoheritage and its importance, as only such awareness triggers activities related to its protection and popularization [20].

The tasks of protection and popularization of geoheritage are closely related to the functions of geoparks. Their complex mechanisms are based on mutual relationships between local natural and cultural resources and between the local community and tourists. As pointed out by Porshnov et al. [30], in the case of the local population, a very important goal of geoparks "...is to strengthen identification of the population with their area and to stimulate the 'pride of place', which in turn produces strong local support for the protection of geological heritage." In turn, geoparks enable tourists and visitors to travel around their territory in order to gain experience, learn, and use geoheritage [95], thereby fulfilling an educational and promotional function.

The current number and spatial distribution of geoparks indicate the difficulty and complexity of the process of their creation. In accordance with the requirements, they should be created through bottom-up initiatives undertaken by local governments and communities, which should be accompanied by political support providing the necessary financial resources [96]. However, as shown by research results reported by, e.g., Farsani et al. 2011 [32], Kubalíková 2019 [34], and Duarte et al. 2020 [35], the interest and involvement of local entities in the process of creation of geoparks is currently insufficient in many cases.

Another difficulty is the fear expressed by stakeholders that, in their opinion, the number, attractiveness, importance, and uniqueness of geotourism resources constituting the basis for creation of geosites are insufficient. The examples described in the literature prove that their attractiveness is often determined by the interesting ideas and creativity of local communities. Good practices that increase the attractiveness of geosites include such geoheritage-based geoproducts as workshops, educational programs, festivals, fairs, local culinary products, handicrafts [95], museums, educational centers, trails, publication of popular literature, maps, and modern communication media [94]. Many of these products present the traditional skills of the inhabitants, which contributes to preservation and better understanding of the local culture, as shown by Farsani et al. [95].

During the creation of geosites as tourist attractions, one should consider their specificity. While most popular attractions are clear and obvious to the viewer, geotourist ones require adequate preparation to facilitate their proper interpretation. Within this group, mixed-use sites, whose appeal derives from the relationship between the geological heritage and its cultural background, prove particularly challenging. Unlike sites presenting clear geological processes and phenomena and relief forms, these rely on recognizing the

uniqueness of the geoheritage and constructing its proper narrative. This is because former industries, crafts, or rock and raw material extraction sites are often extinct, and traces of their existence are only evidenced by buildings and/or their ruins, archival materials, and/or the memory of inhabitants, as indicated by the results of studies conducted by Gordon [14], Dowling [5], Rodriguez et al. [6], Brzezińska-Wójcik, Skowronek [18], and Pijet-Migoń, Migoń [20].

According to the authors, given the complex interdisciplinary context, activities focused on building of knowledge about geosites should be conducted/implemented by scientific and research institutions, as highlighted by Alexandrowicz [2]. They suggest involving interdisciplinary research teams for this purpose. Consequently, in a relatively short time, it would be possible to acquire various source materials, use a variety of methods, tools, and techniques, and prepare holistic documentation of geosites efficiently with a view to creating a product for a wide audience. This is consistent with the expectations of the local government, which, as the results of the study have shown, seeks the support of scientists in the preparation of a set of information on the natural and cultural features of the site.

In the opinion of the authors, the tool aspect is furthermore important. The involvement of scientific and research institutions would result in traditional forms of data acquisition and collection (e.g., in geostationary inventory cards), being supported by new ones–LiDAR, TLS, Unmanned Aerial Vehicle (UAV), among others. Such applications of remote sensing to document geosites have already been presented by numerous authors, e.g., Santos et al. [47], Cayla, Martin [43], and Quesada-Valverde, Quesada-Román [44]. Importantly, this group of state-of-the-art tools and techniques is also already widely used in conservation and preservation activities, as pointed out for example by Calin et al. [46], Rüther [48], and Wei et al. [49].

The model for building the content of geosite presented in the paper is a proposal by the authors to increase efficiency in the process of geoproduct creation. This is important in view of the fact that, in prepared proposals, concerning the establishment of UNESCO geoparks (including, for example, Kamienny Las na Roztoczu Geopark), there must be many such sites, as described by Alexandrowicz [2] and Azman et al. [91].

The authors believe that, irrespective of the success of efforts to achieve the geopark status, well-prepared geosites can be systematically introduced into the tourist space of the region as independent attractions and thus support its development through geotourism.

## 6. Conclusions

Situated on the Polish–Ukrainian border, in the Roztocze region, the village of Huta Różaniecka is an example of a little-known and underdeveloped area that seeks opportunities for its development with reference to its rich natural and cultural heritage. According to the survey, representatives of the local government have recognized its value and acknowledged that it should be protected and promoted. Based on this, it can be concluded that there are emotional ties of the inhabitants (local government) with the place where they live and the heritage that is part of their identity. Such a phenomenon is referred to as a sense of place by Pellow [97] and Agnew et al. [98].

The basis for the creation of the geocultural site at Huta Różaniecka is, on the one hand, the interesting geology and geomorphology expressed in the field by the exposure of the shore of the former Miocene sea with numerous fossils and, on the other hand, the traditions of stonemasonry and its products based on these resources preserved as mining sites and buildings (ruins of the Greek Catholic church).

In its creation, the fundamental difficulty is to show the full context of the site. While its geological and geomorphological features are clear and relatively easy to interpret, the cultural context requires the construction of an appropriate narrative. This is because the site has changed in character: the local crafts have declined, the rock mining sites have become overgrown with forest and fallen into oblivion, and the ethnic and religious structure of the inhabitants has irreversibly changed. The former and current cultural

landscape of the village are different. The creation of a geocultural site in Huta Różaniecka will, therefore, make it possible to evoke the former image and functioning of the village and thus, restore the significance of the preserved heritage elements.

It should therefore be concluded that the idea of creating a geocultural site in Huta Różaniecka, included in the project of the Kamienny Las na Roztoczu Geopark, is correct. Both its spatial dimension, natural features, and cultural context are unique and worth protecting and popularizing. The site has great potential for building a tourism product.

The applied model of proceeding, leading to the construction of the documentation, reveals the wide possibilities of applying modern databases and remote sensing. In gathering and organizing knowledge about the history of the area, the availability of digitized archival materials and cartographic sources is a great help. The use of remote sensing opens up new possibilities of interpretation related to non-invasive methods of data acquisition. In this trend, LiDAR images of the former functioning of the area and spatial models of sites obtained by TLS scanning are interesting and very helpful. In addition to their use in site documentation, their results are increasingly being used in practical terms, e.g., for the preparation of mobile applications and websites (e.g., virtual guides) serving to build and disseminate knowledge about the site.

In addition to the reliability of the message, the most important task in preparing the content of a geocultural site is to keep the audience in mind. This is particularly important given their different perceptual capacities (age, knowledge, education, interests, motivations). Difficult issues need to be communicated to each of these groups in an accessible way and in an attractive form. This conclusion is addressed to those responsible for preparing the final form of the geoproduct. Regardless of whether one succeeds in obtaining the status of a geopark, well-prepared tourist attractions will foster the interest of the viewer and generate tourist traffic.

In such a form of sustainable development as geotourism, it is also important to take into account the opinion of the local community regarding the created tourism product. Thus, further research on the development of the geocultural site project at Huta Różaniecka is needed.

Limitations. The prepared model for creation of the content of the geocultural site in Huta Różaniecka is based on the characteristic traits of this site and area. Even though it is of regional importance, it is distinguished by its authenticity. It contributes to the maintenance of craft traditions dating back to the 19th century and based on local rocks and to popularization of the knowledge of the non-existent multicultural past. The process of creation of the context of each geocultural site should consider its individual features. Therefore, all available data sources related to geology, palaeontology, archaeology, history, anthropology, culture, and ecology should be taken into account and verified. Moreover, their proper interpretation is necessary.

**Author Contributions:** Conceptualization, E.S., T.B.-W. and W.K.; methodology, T.B.-W., E.S. and W.K.; software, W.K.; validation, E.S., T.B.-W. and W.K.; formal analysis, E.S., T.B.-W. and W.K.; investigation, E.S., T.B.-W. and W.K.; resources, E.S., T.B.-W. and W.K.; data curation, E.S., T.B.-W. and W.K.; writing—original draft preparation, E.S., T.B.-W. and W.K.; writing—review and editing, E.S., T.B.-W. and W.K.; visualization, W.K. and T.B.-W.; supervision, E.S., T.B.-W. and W.K.; project administration, E.S., T.B.-W. All authors have read and agreed to the published version of the manuscript.

**Funding:** This research received no external funding.

**Institutional Review Board Statement:** Not applicable.

**Informed Consent Statement:** Informed consent was obtained from all subjects involved in the study.

**Data Availability Statement:** Data is contained within the article.

**Acknowledgments:** We are grateful to the anonymous reviewers for taking the trouble to review the article and for their constructive comments. We would also like to thank Marcin Mazurek for photographing the ruins of the church from a drone, Marcin Materniak for the replica of the 1896

engraving of the Huta Różaniecka church, and Bogdan Skibiński for his cooperation during the field inventory.

**Conflicts of Interest:** The authors declare no conflicts of interest.

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
