# Peer review of "How to Create a Geocultural Site’s Content–Huta Różaniecka Case Study (SE Poland)"

_sustainability, doi:10.3390/su16052193_

Round 1

Reviewer 1 Report

Comments and Suggestions for Authors

The subject of geoparks and the valorization of geosites is an issue that is increasingly being studied, given the heritage importance of these UNESCO-listed territories and their growing demand from visitors.

The research presented here points to interesting contributions within this spectrum of inventory, preservation and enhancement, which are recognized as ways of qualifying these geoparks.

In this sense, the methodology requires further development and detailing of the research design, the tools used, the reasons for their suitability and the extrapolation of the results. It would be important to include infograms or diagrams of the research carried out, as well as the media used to collect the data.

The figures need to be checked, improved and their appropriate framing in the text promoted.

The bibliographical citations throughout the text should be checked so that they comply with the rules of the publication and their formatting.

Check recent studies on the problem under study, which have contributed to the soundness of the research, updating the references and deepening the literature review. The discussion of the concepts of geocultural and ecocultural would be of value to the research. It would be interesting to verify or compare different practices, or rather good practices, that have been developed in geoaprks in this context of valuing georesources.

Comments on the Quality of English Language

Minor editing of English language required

Author Response

We would like to thank you for your careful consideration of the issues addressed in the refereed research. All of the Reviewer's valuable comments and suggestions have been addressed in the revised version of the manuscript.

A description of the methodology used has been supplemented. A flowchart (new Figure 2) has been added presenting the whole research process leading to the extraction of the geoculture site.

Figures have been completed. New ‘Figure 2’ (flowchart) added and geoculture site coverage added to figure x. References to all figures in the text were reviewed and verified.

Bibliographic citations throughout the text have been completed and existing citations revised in accordance with MDPI home style rules (also in the final list of references).

The state of the art has been updated. The literature review has been completed. The geocultural context was expanded by adding references to the achievements of authors in the field of geoarchaeology. Following this thread, the earliest history of the study area was added in subchapter 4.2. The ‘Discussion’ chapter has been expanded by studies of the mechanisms of geoparks functioning and a review of good practices affecting their attractiveness.

English spelling was checked and verified throughout the manuscript, with minor corrections made. 

Reviewer 2 Report

Comments and Suggestions for Authors

1. Working upon a specific territory, developing a geocultural site, in order to improve geotourism and a generalized notion (specific among local governments) that geosites and culture can add value to educational and recreation purposes (including via tourism activity).

2. It contemplates the aggregation of both geo-natural and cultural resources, combining them in the same reality (geocultural site), which is a nice option considering the narrowed profile of geotourism consumers.

3. First, it contributes to the protection of two different types of resources (geological and cultural), which is strongly important to encourage their sustainability.

Secondly, it works upon a concept of attraction that not only serves for fruition, but also can work aligned with educational and pedagogical approaches, which are also very important to maintain and preserve past evidence (local habitants will never protect what they do not know or understand).

Thirdly, combines the use of rock, via quarries and construction, interlinking the natural resources availability with what humans can do with those resources.

However, authors should consider improving their future approach by focusing on two areas that may enrich the geocultural site's narrative: Archaeology and Anthropology. The first one, perhaps by adding into the discourse a wider lifespan of local landscape occupation, which not only will show how the area was habited since long ago, but also how important it was. Several archaeological evidence all over the world show how local communities, since Prehistory, started producing tools and used rocks to construct artificial settlements (once abandoning caves and natural shelters). The second one (Anthropology), by demonstrating how geo-natural sites were and are present in daily lifestyles of local populations, creating both material and immaterial forms of cultural heritage linked to a common natural resource exploitation or use. In many cases, for example, the presence of toponymy, legends, beliefs, or other forms of expression can record this relation.

4. Personally, I would like to see the geocultural site clearly mapped and bounded, since there is always a need to create frontiers of our study object. From that, different kinds of maps could be prepared and used to improve local trails and paths, adding clear and fundamental information about the geocultural site itself.

The presentation of the interviews was absolutely and extremely compiled, perhaps too much, probably because of space required for the remaining text. Personally, I would consider it important to see 8at least) the question guide.

5. Overall, I will underline two main aspects:

- the author's reference about the local governments perception of what this kind of research and (geocultural) creation can add to a poorly developed territory;

- and the limitations derived from nuances related to the meaning of a cultural reality, whose perception is time changing.

However, I am not that sure about the tourism potential of Huta Rózaniecka geocultural site, at least, in the same dimension defended in the conclusions by the authors. It is worth noting, however, that right from the start, authors realized that combining geology and geomorphological aspects with cultural attractions is strategically more strong that working solely upon geo-natural aspects.

7. I already noted that this geocultural site could be well defined by a mapped area, which should include the referred attractions and specific boundaries (even being part of a wider reality (Kamienny Las na Roztoczu Geopark).

In general, images are well presented, including the indication of North and scale (when mapping) and with good definition.

Please, consider the following two suggestions:

- assume a boundary of territory as part of the geocultural site authors are working on, preferably mapping all the different realities that form it;

- seeing that landscape as an open entity (also deriving from the use of human beings through time), think about involving Archaeology and archaeologists point of view; it is worth nothing that some of the areas can register the presence of past communities, and this would definitely enrich your cultural discourse.

Comments on the Quality of English Language

Very small changes, easily detected by an English native speaker

Author Response

Thank you very much for reviewing our manuscript, as well as for your constructive feedback/comments, which we have tried to take into account.

General comments 1-3: Thank you for this extremely apt synthetic summary of the research problem undertaken. We fully agree with this summary diagnosis.

Thank you for pointing out archaeology and anthropology, which enrich the geocultural site's narrative. Following this suggestion, in the ‘Introduction’ chapter, we have expanded the geocultural context by supplementing it with publications in geoarchaeology. In addition, in subsection 4.2 we supplemented the prehistoric and earliest history of the study area. However, due to the lack of scientific studies in Anthropology for the study area, we were unable to expand the cultural context.

Following up on the Reviewer's suggestion (items 4 and 7) to map and delimit the geocultural site, a signature of the geocultural site boundary was added to ‘Figure 7’ showing the geology of the study area. Unfortunately, the Stone Forest Geopark in Roztocze was not depicted, due to its area which is beyond the scale of the detailed maps, and at the same time too small to be clearly visible against the background of the entire Roztocze region (Figure 1).

In the ‘Materials and methods’ chapter, a description of the survey methodology was supplemented with 10 questions from the interview questionnaire.

English spelling was checked and verified throughout the manuscript, with minor corrections made.